# ACCELERATING FIRST ORDER OPTIMIZATION ALGORITHMS

## ABSTRACT

There exist several stochastic optimization algorithms. However in most cases, it is difficult to tell for a particular problem which will be the best optimizer to choose as each of them are good. Thus, we present a simple and intuitive technique, when applied to first order optimization algorithms, is able to improve the speed of convergence and reaches a better minimum for the loss function compared to the original algorithms. The proposed solution modifies the update rule, based on the variation of the direction of the gradient during training. We conducted several tests with Adam and AMSGrad on two different datasets. The preliminary results show that the proposed technique improves the performance of existing optimization algorithms and works well in practice.

## 1 INTRODUCTION

There are several ways to improve learning in neural networks such as improving the architecture, finding the optimal parameters, playing with the data representation, choosing the best optimization algorithm etc. In this paper, we are interested in the gradient-descent-based method for optimizing the learning process in neural network architectures. Optimization algorithms in machine learning (especially in neural networks) aim at minimizing an objective function (generally called loss or cost function), which is intuitively the difference between the predicted data and the expected values. The minimization consists of finding the set of parameters (weights) of the architecture that give best results in the targeted tasks such as classification, prediction or clustering. The problem of finding a set of weights to minimize residuals in a feedfoward neural network is not a trivial one. It is nonlinear and dynamic in that any change of one weight requires adjustment of many others (Eberhart & Kennedy, 1995).

Several optimization algorithms exist in the literature. The majority of them are first order methods (as they use first derivatives of the function to minimize), and are based on the gradient descent. Gradient descent techniques, e.g. back-propagation of error, are used to find a matrix of weights that meets the error criterion. Adam (Adaptive Moment estimation) is probably the most popular optimization algorithm in the literature (Goodfellow et al., 2016; Isola et al., 2017; Xu et al., 2015). However, it has been recently proved that Adam (even discounting the fact that its convergence proof has some issues), is unable to converge to the optimal solution for a simple convex optimization setting (Reddi et al., 2018). A more recent algorithm (AMSGrad) fixes the problems of Adam and others stochastic optimization methods by endowing them with a long-term memory. Unfortunately, Adam can still outperforms AMSGrad in some cases[1] as we will see in our experiments. This suggests that, there is no a better optimizer than others as each can be good than others or vice versa depending on the problem. Thus, we investigate a new technique that improve thee empirical performance of any first order optimization algorithm, while preserving their property. We will refer to the solution when applied to an existing algorithm $A$ as $AA$ for (Accelerated Algorithm). For example, for AMSGrad and Adam, the modified versions will be AAMSGrad and AAdam. The proposed solution improves the convergence of the original algorithm and finds a better minimum faster. Our solution is based on the variation of the direction of the gradient with respect to the previous. We conducted several tests on problems where the shape of the loss is simple (has a convex form) like Logistic regression, but also with non trivial neural network architectures such as deep

---

1. https ://fdlm.github.io/post/amsgrad/

Convolutional Neural Networks. We used *MNIST* [2] and *Movie Review Data* [3] datasets to validate our solution. Note that we limited our experiments to SGD, Adam and AMSGrad, but the solution can be applied to others algorithms as well. The preliminary results suggests that adding the proposed solution to the update rule of a given optimizer makes it performs better.

The rest of the paper is organized as follows : Section 2 presents some related work ; Section 3 describes the proposed solution and provides its theoretical analysis ; Section 4 presents the experiments setup, the results and discussions about the performance assessment of the proposed solution and its possible integration in other optimizers. The paper ends with some concluding remarks.

## 2 Related Work

Learning in neural networks is done by minimizing an error function. This function therefore measures the difference between the expected and the computed outputs on the complete sample. An error close to 0 implies that the network correctly predicts the expected outputs of the data on which it has learned. Minimization of the loss involves finding the values of a set of parameters (weights) that minimizes the function (this minimum could be local or global). What makes the problem more complex is that, the general shape of the loss function is very poorly understood. Logistic regression (LR) and softmax (multiclass generalization of LR) do not belong to that class as they have well-studied convex objectives (Rennie, 2005). However that is not the case for neural networks with more than one layer where the shape of the loss is generally neither convex nor concave and thus can admit several minima (Choromanska et al., 2015). The loss function is usually minimized using some form of stochastic gradient descent (SGD) (Bottou, 2010), in which the gradient is evaluated using the back-propagation procedure (LeCun et al., 1989). Gradient Descent is one of the most popular algorithms to perform such a task and is the much more common way to optimize neural networks (Ruder, 2016). To be used in back-propagation, the loss function must satisfy some properties such as being able to be written as an average, and not be dependent on any activation values of a neural network besides the output values. There exist many loss functions but the quadratic or mean squared error and the cross-entropy are the most commonly used in the domain.

### 2.1 How Gradient Descent Method Works ?

Let J($\theta$) be a function parameterized by a model's parameters $\theta \in \mathbb{R}^n$, sufficiently differentiable of which one seeks a minimum. The gradient method builds a sequence that should in principle approach the minimum. For this, we start from any value $x_0$ (a random value for example) and we construct the recurrent sequence by :

$$\theta_{n+1} = \theta_n - \eta \cdot \nabla_{\theta_n} J(\theta) \tag{1}$$

where $\eta$ is the learning rate. For adaptive method like Adam, the learning rate is variable for each parameters. This method is ensured to converge, even if the input sample is not linearly separable, to a minimum of the error function for a well-chosen learning rate. There exist several variants of this method : there are first-order and second-order methods. While first order methods use first derivatives of the function to minimize, second order methods make use of the estimation of the Hessian matrix (second derivative matrix of the loss function with respect to its parameters) (Schaul et al., 2013; LeCun et al., 1993). The latter determines the optimal learning rates (or step size) to take for quadratic problems. While such approach provides additional information useful for optimization, computing accurate second order derivatives is too computationally expensive for large models and the value computed is usually a bad approximation of the Hessian. In this paper, we will only focus on first order techniques which are the most popular in machine learning domain and are most suited for large scale models.

Adam (Kingma & Ba, 2014) is a first-order-gradient based algorithm of stochastic objective functions, based on adaptive estimates of lower-order moments. The first moment normalized by the second moment gives the direction of the update. Adam updates are directly estimated using a running average of first and second moment of the gradient. It computes adaptive learning rates for each parameter. In addition to storing an exponentially decaying average of past squared gradients

---

2. http ://yann.lecun.com/exdb/mnist/
3. http ://www.cs.cornell.edu/people/pabo/movie-review-data/

$v_t$, Adam also keeps an exponentially decaying average of past gradients $m_t$, similar to momentum. It has two main components : a momentum component and an adaptive learning rate component. It can be viewed as a combination of RMSprop (Hinton et al.)) and momentum techniques NAG (Nesterov, 1983). Adam has a bias-correction feature which helps it slightly outperforms previous adaptive learning rate methods towards the end of optimization as gradients become sparser (Ruder, 2016). To our knowledge, Adam is one of the latest state of the art optimization algorithms being used by many practitioners of machine learning. There exist several techniques to improve Adam such as fixing the weight decay (Loshchilov & Hutter, 2017), using the sign of the gradient in distributing learning cases (Bernstein et al., 2018), switching from Adam to SGD (Keskar & Socher, 2017). However, it has been recently proved that Adam, beyond the fact that the convergence proof has some mistakes, is unable to converge to the optimal solution for a simple convex optimization setting (Reddi et al., 2018). A more recent algorithm (AMSGrad) fixes the problems of Adam and others stochastic optimization methods by endowing them with a long-term memory. The main difference between AMSGrad and ADAM is that, AMSGrad maintains the maximum of all $v_t$ (exponentially decaying average of past squared gradients) until the present time step and uses this maximum value for normalizing the running average of the gradient. AMSGrad either performs similarly, or better, than Adam on some commonly used problems in machine learning. The update rule is as follows :

$$
\begin{aligned}
\theta_{n+1} &= \theta_n - \frac{\alpha}{\sqrt{\hat{v}_n + \epsilon}} \hat{m}_n \\
m_n &= \beta_1 \cdot m_{n-1} + (1 - \beta_1) \cdot \nabla_{\theta_n} J(\theta) \\
v_n &= \beta_2 \cdot v_{n-1} + (1 - \beta_2) \cdot \nabla_{\theta_n} J(\theta)^2 \\
\hat{m}_n &= \frac{m_n}{1 - \beta_1^n} \\
\hat{v}_n &= max(\hat{v}_{n-1}, \frac{v_n}{1 - \beta_2^n}).
\end{aligned}
$$

Recall that the aim of an optimizer is to finds parameters that minimize a function, knowing that we don't really have any knowledge on how the function looks like. If one knew the shape of the function to minimize, it would be easy to take accurate steps that lead to the minimum. We don't know how the shape is but, each time we take a step (using any of the optimizers) we can know if we passed a minimum by computing the product of the current and the past gradient, and check if it is negative. This informs us on the curvature of the loss surface and as far as we know, it is the only accurate information we have on the curvature of our loss function in real time. The proposed method exploits this knowledge to improve the convergence of an optimizer. There exist several ways to improves each optimizers such as using momentum techniques etc. However, none of them used the variation of the direction of the gradient as an information to compute the next step. The proposing solution is not specific to a particular optimizer as many solutions that have been proposed. It can be applied even on an already proposed accelerated version of the algorithm which makes it easily adaptable on every optimizers.

## 3 A METHOD TO ACCELERATE FIRST ORDER OPTIMIZATION ALGORITHMS

### 3.1 NOTATION.

$\|x_i\|_2$ denotes the $l_2$-norm of $i^{th}$ row of x. The projection operation $\prod_{\mathcal{F},A}(x)$ for A $\in \mathcal{S}_+^d$ (the set of all positive definite $d * d$ matrices) is defined as arg $min_{x \in \mathcal{F}} \|A^{1/2}(x - y)\|$ for $y \in \mathbb{R}^d$. F has bounded diameter $D_\infty$ if $\|x - y\|_\infty \leq D_\infty$ forall $x, y \in \mathcal{F}$. All operations applied on vectors or matrices are element-wise.

### 3.2 INTUITION AND PSEUDO-CODE.

Here we present the intuition behind the proposed solution. See algorithm 1 for pseudo-code of our proposed method applied to the generic adaptive method setup proposed by Reddi et al. (2018). For Adam, $\varphi_t(g_1, ..., g_t) = (1 - \beta_1) \sum_{i=1}^t \beta_1^{t-i} g_i$ and $\psi_t(g_1, ..., g_t) = (1 - \beta_2) diag\left(\sum_{i=1}^t \beta_2^{t-i} g_i^2\right)$.

---

**Algorithm 1** *Accelerated - Generic Adaptive Method Setup*

---

**Input :** $x_1 \in F$, step size $(\alpha_t > 0)_{t=1}^T$, sequence of functions $(\varphi_t, \psi_t)_{t=1}^T$
$t \leftarrow 0$
$pg_0 \leftarrow 0$ (Initialize previous gradient).
**repeat**
$\quad g_t = \nabla f_t(x_t)$
$\quad V_t = \psi_t(g_t)$
$\quad$**if** $g_t * pg_t < 0$ **then** $m_t = \varphi_t(g_t)$
$\quad$**else** : $m_t = \varphi_t(max(g_t, pg_t))$
$\quad \hat{x}_{t+1} = x_t - \alpha_t m_t V_t^{-1/2}$
$\quad x_{t+1} = \prod_{F,\sqrt{V_t}}(\hat{x}_{t+1})$
$\quad pg_{t+1} = g_t$
$\quad t \leftarrow t + 1$
**until** t > T

---

To give an odd entrance to our method, lets take the famous example of the ball rolling down a hill. If we consider that our objective is to bring that ball (parameters of our model) to a lowest elevation of a road (cost function), what we do is to push the ball with a force equal to the max of the past (t-1) and the current computed force. This is done by taking the maximum between the computed gradient (taken by any optimizer) and the previous gradient. The ball will gain more speed as it continues to go in the same direction and looses its current speed as soon as it passes over a minimum. Note that, the previous gradient can be replace by $\tan\left(\frac{\arctan(pg_t) + \arctan(g_t)}{2}\right)$ which is the angle between the two gradients (sum of vectors). This will reduce the step taken if we consider the maximum and will still converge faster but slower. Another solution could be to stop accelerating the ball and let the original optimizer takes the full control of the rest, once the direction change for the first time. This solution also works. We will only focus on the case where the gradient is replaced by the maximum between its current value and the past value. Note that $V_t$ do not change. The change we make to the optimizer do not alterate its convergence since the quantity $R$ which essentially measures the change in the inverse of learning rate of the adaptive method with respect to time keep its value ($v_t remain unchanged$ and the current gradient will not be more than the previous one). We provided in the appendix, a theoretical proof that the method we are proposing do not change the bound of the regret.

## 3.3 CONVERGENCE ANALYSIS

We assume that if we are able to prove that modifying one optimizer with the method do not alter its convergence, then the same applies for other optimizers as well. Thus, we analyze the convergence of AAMSGrad using the online learning framework.

**Theorem 1** *Assume that $\mathcal{F}$ has bounded diameter $D_\infty$ and*

$$\|\nabla f_t(x)\|_\infty \leq G_\infty$$

*for all $t \in [T]$ and $x \in \mathcal{F}$. With $\alpha_t = \frac{\alpha}{\sqrt{T}}$, AAMSGrad achieves the following gurantee, for all $T \geq 1$ :*

$$
\begin{aligned}
R_T \leq &\frac{D_\infty^2 \sqrt{T}}{2\alpha(1-\beta_1)} \sum_{i=1}^d (\hat{v}_{T,i}^{-1/2} + \frac{\hat{v}_{2,i}^{-1/2}}{\alpha_2}) \\
&+ \frac{D_\infty^2}{2(1-\beta_1)} \sum_{t=1}^T \sum_{i=1}^d \frac{\beta_{1t} \hat{v}_{t,i}^{1/2}}{\alpha_t} \\
&+ \frac{\alpha\sqrt{1 + log(T)}}{(1-\beta_1)^2(1-\gamma)\sqrt{(1-\beta_2)}} \sum_{i=1}^d \|g_{1:T,i}\|_2
\end{aligned}
\tag{2}
$$

The proof of this bound is given in the appendix. The following result falls as an immediate corollary of the above result :

**Corollary 1.1** *Let $\beta_{1t} = \beta_1 \lambda^{t-1}, \lambda \in (0,1)$ in Theorem 1, then we have :*

$$
\begin{aligned}
R_T \leq {} & \frac{D_\infty^2 \sqrt{T}}{2\alpha(1-\beta_1)} \sum_{i=1}^{d} (\hat{v}_{T,i}^{-1/2} + \frac{\hat{v}_{2,i}^{-1/2}}{\alpha_2}) \\
& + \frac{\beta_1 D_\infty^2 G_\infty}{2(1-\beta_1)(1-\lambda)^2} \\
& + \frac{\alpha\sqrt{1+log(T)}}{(1-\beta_1)^2(1-\gamma)\sqrt{(1-\beta_2)}} \sum_{i=1}^{d} \|g_{1:T,i}\|_2
\end{aligned}
\tag{3}
$$

The regret of $AAMSGRAD$ can be bounded by $O(G_\infty \sqrt{T})$. The term $\sum_{i=1}^{d} \|g_{1:T,i}\|_2$ can also be bound by $O(G_\infty \sqrt{T})$ since $\sum_{i=1}^{d} \|g_{1:T,i}\|_2 << dG_\infty \sqrt{T}$ (Kingma & Ba, 2014).

## 4 EXPERIMENTS

In order to empirically evaluate the proposed solution, we investigated different models : Logistic Regression and Convolutional Neural Networks (CNN). We used the same parameter initialization. $\beta_1$ was set to 0.9 and $\beta_2$ was set to 0.999 as recommended for Adam (Kingma & Ba, 2014) and the learning rate was set to 0.001 and 0.01. The algorithms were coded using Googles TensorFlow [4] API and the experiments were done using the built-in TensorFlow models, making only small edits to the default settings.

### 4.1 IMAGE RECOGNITION

No pre-processing was applied on MNIST images. All optimizers were trained with a mini-batch size of 100. All weights were initialized from values truncated using normal distribution with standard deviation 0.1. The biases values were initialized to 0.1. We used the softmax cross entropy as the loss function.

Logistic regression has a well-studied convex objective, making it suitable for comparison of different optimizers without worrying about local minimum issues (Kingma & Ba, 2014). We implemented the same model used for comparing Adam with other optimizers in the original paper on Adam. In Figure 1 , we see that the accelerated versions outperform the original algorithms (SGD and ADAM) from the beginning to the end of the training even when we change the learning rate. Our intuition that the method is able to improve the convergence and finds a better minimum is thus consistent. CNNs (LeCun et al., 1995) are neural networks with layers representing convolving filters (Krizhevsky et al., 2012) applied to local features. CNNs are now used in almost every task in machine learning (i.e. text classification (Tato et al., 2017), speech analysis (Abdel-Hamid et al., 2014), etc.). Figures 2 shows the results of running a neural architecture with one convolutional layer and one fully connected layer on MNIST data with the learning rate set to 1E-2. Again, the accelerated versions outperform the originals ( Adam and AMSGrad). We can also note that, in this example, Adam outperforms AAMSGrad which suggests that the performance of an optimizer depends on the data (shape of the loss).

### 4.2 MOVIE REVIEWS DATA

We experimentally evaluated our method on movie reviews data. We ran a convolutional model on *sentence polarity dataset v1.0* which consists of 5331 positive and 5331 negative processed sentences / snippets. We used a slightly simplified version [5] of the CNN proposed by (Kim, 2014) on the dataset. All the parameter setting of the models was left unchanged. The dimensionality of character embedding was set to 128, the filter sizes was set to 3,4,5, the number of filters per filter size

---

4. https ://www.tensorflow.org/
5. https ://github.com/dennybritz/cnn-text-classification-tf

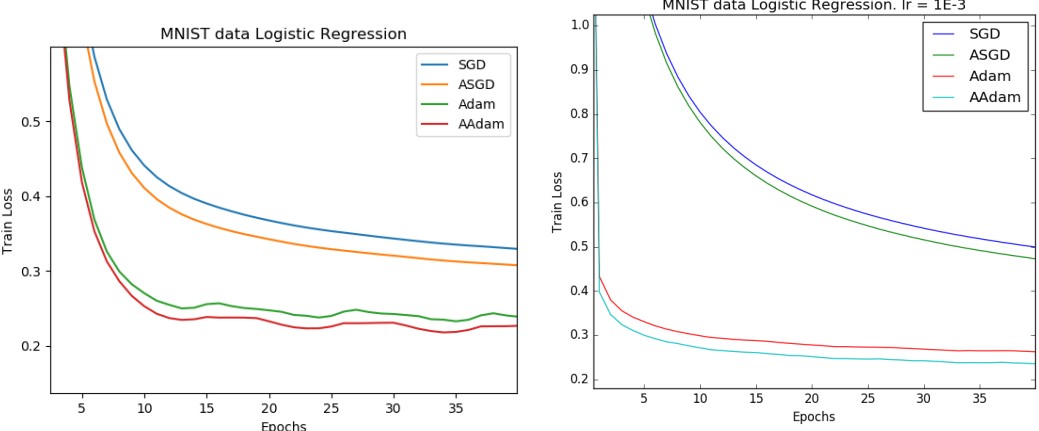

FIGURE 1 – Logistic regression training negative log likelihood on MNIST with learning rate = 1E-2 (first plot on the left) and learning rate = 1E-3 (second plot).

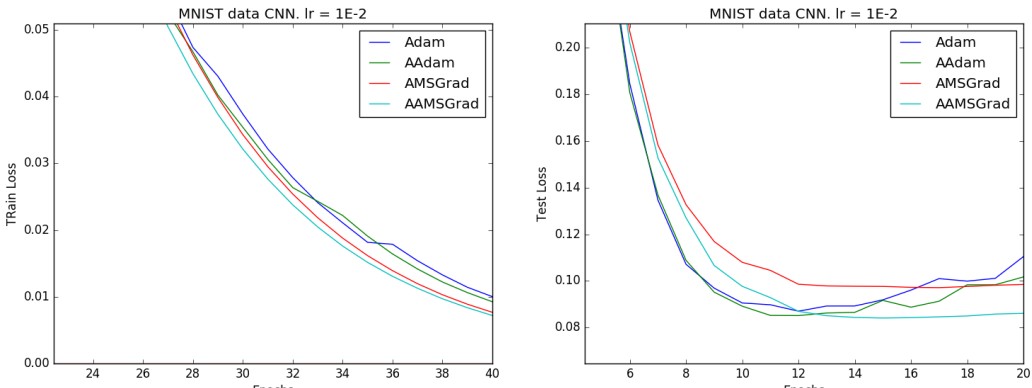

FIGURE 2 – Convolutional Neural Networl with on conv and one fully connected layer on MNIST. Learning rate = 1E-2 on training step (first plot on the left) and testing step (second plot).

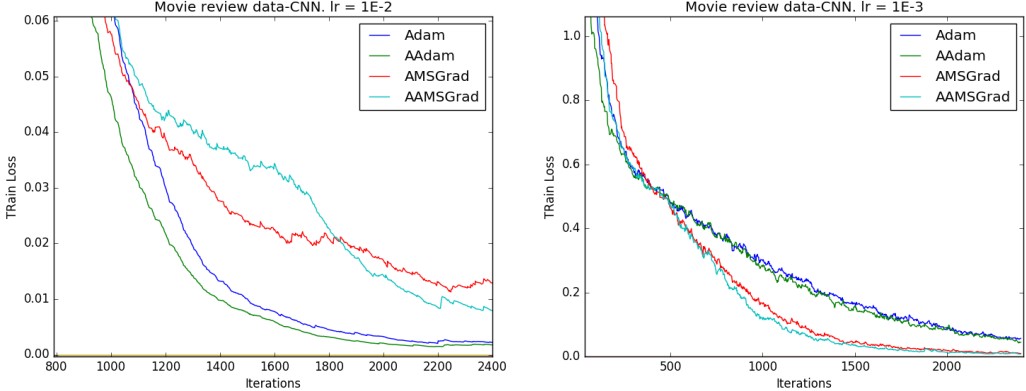

FIGURE 3 – Convolutional Neural Network on movie review data. Learning rate = 1E-2 (plot on the left) and Learning rate = 1E-3 (seconf plot) on training step.

was set to 128, the dropout keep probability set to 0.5, the batch size was set to 64, and the number of training epochs was set to 50 ( Results are presented in Figure 3. As we expected, AAdam consistently outperforms Adam on both cases (lr=1E-2 and 1E-3) and thus converges faster. AAMSGrad outperforms AMSGrad and finds a better minimum also on both cases.

## 5    CONCLUSION AND FINAL REMARKS

In this paper, we have presented a simple and intuitive method that modifies any optimizer update rule to improve its convergence. There is no optimizer that works for all problems. Each optimizer has its advantages and disadvantages. No one can tell in advance which will be the best choice. The solution we are proposing is to speed up the convergence of any optimizer (as each is important) based on the variation of the sign of the gradient. Instead of using the gradient as it is for computing the next step, we used the maximum of the past and the current gradient to compute the step if both has the same sign (the direction did not change) else we keep the current gradient. We conducted a successful preliminary experiment with three well known optimizers (SGD, Adam and AMSGrad) on different models using two datasets (MNIST and movie review data). In all our experiments, the accelerated versions outperformed the originals. In worst cases, both have the same convergence which suggests that the accelerated versions are at least as good as the originals. This work takes those optimizers one step further, and improves their convergence without noticeably increasing complexity. The only drawback of the proposed solution is that it takes a little bit more computation than the standard approach (as it has to cumpute the if else instruction). The new update rule depends only on the variation of the direction of the gradient, which means that it can be used in any other optimizer for the same goal.

### ACKNOWLEDGMENTS

The authors would like to thank the anonymous referees for their valuable comments and helpful suggestions. The work is supported by the GS501100001809Natural Sciences and Engineering Research Council of Canadahttp ://www.nserc-crsng.gc.ca/index$_e ng.aspDiscoveryGrantProgram$.

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

## 6 APPENDIX : PROOF OF THE CONVERGENCE OF AAMSGRAD

The proof presented below is along the lines of the Theorem 4 in Sashank J et al (ICLR-2018), which provides a proof of convergence for AMSGrad.

The goal is to prove that, the regret of AAMSGrad is bounded by :

$$
R_T \leq \frac{D_\infty^2}{2(1-\beta_1)} \sum_{i=1}^{d} (\frac{\hat{v}_{T,i}^{-1/2}}{\alpha_T} + \frac{\hat{v}_{2,i}^{-1/2}}{\alpha_2}) + \frac{D_\infty^2}{2(1-\beta_1)} \sum_{t=1}^{T} \sum_{i=1}^{d} \frac{\beta_{1t} \hat{v}_{t,i}^{1/2}}{\alpha_t}
$$
$$
+ \frac{\alpha \sqrt{1+log(T)}}{(1-\beta_1)^2(1-\gamma)\sqrt{(1-\beta_2)}} \sum_{i=1}^{d} \|g_{1:T,i}\|_2
\tag{4}
$$

For each optimization method, we have :

$$
x_{t+1} = x_t - \alpha U
\tag{5}
$$

Where $\alpha$ is the step size. Note that, the value of the update U is the gradient (or slope) of a line. For example, this line is the slope of the tangent of the loss at $x_t$ in SGD.

AAMSGrad has 2 rules for the update which differ depending on whether the direction of the update changes or not. The current step taken by AAMSGrad is :

$$
(1) \ x_{t+1} = \Pi_{F,\sqrt{\hat{V}_t}}(x_t - \alpha_t m_t/\sqrt{\hat{v}_t}) \text{ if sign(d) has changed.}
$$
$$
(2) \ x_{t+1} = \Pi_{F,\sqrt{\hat{V}_t}}(x_t - \alpha_t m_x/\sqrt{\hat{v}_t} \text{ otherwise}
$$
$$
\leq \Pi_{F,\sqrt{\hat{V}_t}}(x_t - max(m_t/\sqrt{\hat{v}_t}, m_{t-1}/\sqrt{\hat{v}_{t-1}}))
$$
$$
m_x = \beta_1 \cdot m_{t-1} + (1-\beta_1) \cdot max(\nabla_{\theta_t} J(\theta), \nabla_{\theta_{t-1}} J(\theta))
\tag{6}
$$

All operations are element wise. When the direction of the current update changes from the past one, the current update is the same as in AMSGrad (rule (1)). When the direction stays the same, the update is the maximum between the update that AMSGrad would have taken and the previous update it took (rule (2)). Thus the regret bound of AAMSGrad is :

$$
R_T \leq max(R_{T(1)}, R_{T(2)})
\tag{7}
$$

Where $R_{T(1)}$ is the regret when we consider only the first update rule of AAMSGrad. Please note that, the bound of $R_{T(1)}$ is similar to that of AMSGrad. $R_{T(2)}$ is the regret if we consider only the second rule. The proof will consist of finding the bound for $R_{T(2)}$. The second update rule of AAMSGrad can be rewritten as follows :

$$
x_{t+1} \leq \Pi_{F,\sqrt{\hat{V}_t}}(x_t - U) \ where \ U = max(\alpha_t \hat{V}_t^{-1/2} m_t, \alpha_{t-1} \hat{V}_{t-1}^{-1/2} m_{t-1})
\tag{8}
$$

We will only focus on the case where U is $\alpha_{t-1} \hat{V}_{t-1}^{-1/2} m_{t-1}$ as for the case where U is $\alpha_t \hat{V}_t^{-1/2} m_t$, it is AMSGrad.

We begin with the following observation :

$$
x_{t+2} \leq \Pi_{F,\sqrt{\hat{V}_t}}(\alpha_t \hat{V}_t^{-1/2} m_t) = \min_{x \epsilon F} \left\| \hat{V}_t^{-1/4}(x - (x_t - \alpha_t \hat{V}_t^{-1/2} m_t)) \right\|.
\tag{9}
$$

Using Lemma 4 of Sashank J et Al (2018) and along the lines of their proof of Theorem 4, we have the following :

$$
\left\| \hat{V}_t^{-1/4}(x_{t+2} - x^*) \right\|^2 \leq \left\| \hat{V}_t^{-1/4}(x_t - \alpha_t \hat{V}_t^{-1/2} m_t - x^*) \right\|^2
$$
$$
\leq \left\| \hat{V}_t^{-1/4}(x_t - x^*) \right\|^2 + \alpha_t^2 \left\| \hat{V}_t^{-1/4} m_t \right\|^2 - 2\alpha_t \langle \beta_{1,t} m_{t-1} + (1-\beta_{1,t})g_t), x_t - x^* \rangle
\tag{10}
$$

Re-arranging the equation 10, and applying Cauchy-Schwarz and Young's inequality, we have :

$$\langle g_t, x_t - x^* \rangle \le \frac{1}{2\alpha_t(1-\beta_{1t})} \left[ \left\| \hat{V}_t^{-1/4}(x_t - x^*) \right\|^2 - \left\| \hat{V}_t^{-1/4}(x_{t+2} - x^*) \right\|^2 \right] + \frac{\alpha_t}{2(1-\beta_{1t})} \left\| \hat{V}_t^{-1/4} m_t \right\|^2$$
$$\frac{\beta_{1t}\alpha_t}{2(1-\beta_{1t})} \left\| \hat{V}_t^{-1/4} m_{t-1} \right\|^2 + \frac{\beta_{1t}}{2\alpha_t(1-\beta_{1t})} \left\| \hat{V}_t^{1/4}(x_t - x^*) \right\|^2$$

$$(11)$$

Knowing that a convex and differentiable function $f_t$ (the loss) can be lower bounded by a hyperplane at its tangent (Lemma 10.2 in Kingma & Ba, 2015), we have :

$$R_{T(2)} = \sum_{t=1}^{T} f_t(x_t) - f_t(x^*) \le \sum_{t=1}^{T} \langle g_t, x_t - x^* \rangle$$

$$\le \sum_{t=1}^{T} \left[ \frac{\left\| \hat{V}_t^{-1/4}(x_t - x^*) \right\|^2 - \left\| \hat{V}_t^{-1/4}(x_{t+2} - x^*) \right\|^2}{2\alpha_t(1-\beta_{1,t})} + \frac{\alpha_t}{2(1-\beta_{1t})} \left\| \hat{V}_t^{-1/4} m_t \right\|^2 \right.$$

$$\left. + \frac{\beta_{1t}\alpha_t}{2(1-\beta_{1t})} \left\| \hat{V}_t^{-1/4} m_{t-1} \right\|^2 + \frac{\beta_{1t}}{2\alpha_t(1-\beta_{1t})} \left\| \hat{V}_t^{-1/4}(x_t - x^*) \right\|^2 \right]$$

$$\le \sum_{t=1}^{T} \left[ \frac{\left\| \hat{V}_t^{-1/4}(x_t - x^*) \right\|^2 - \left\| \hat{V}_t^{-1/4}(x_{t+2} - x^*) \right\|^2}{2\alpha_t(1-\beta_{1,t})} + \frac{\beta_{1t}}{2\alpha_t(1-\beta_{1t})} \left\| \hat{V}_t^{-1/4}(x_t - x^*) \right\|^2 \right] +$$

$$\frac{\alpha\sqrt{1+log(T)}}{(1-\beta_1)^2(1-\gamma)\sqrt{(1-\beta_2)}} \sum_{i=1}^{d} \|g_{1:T,i}\|_2$$

$$\le \frac{1}{2(1-\beta_1)} \sum_{t=1}^{2} \frac{\left\| \hat{V}_t^{-1/4}(x_t - x^*) \right\|^2}{\alpha_t} + \frac{1}{2(1-\beta_1)} \sum_{t=3}^{T} \left[ \frac{\left\| \hat{V}_t^{-1/4}(x_t - x^*) \right\|^2}{\alpha_t} - \frac{\left\| \hat{V}_{t-2}^{-1/4}(x_t - x^*) \right\|^2}{\alpha_{t-2}} \right]$$

$$+ \sum_{t=1}^{T} \frac{\beta_{1t}}{2\alpha_t(1-\beta_{1t})} \left\| \hat{V}_t^{-1/4}(x_t - x^*) \right\|^2 + \frac{\alpha\sqrt{1+log(T)}}{(1-\beta_1)^2(1-\gamma)\sqrt{(1-\beta_2)}} \sum_{i=1}^{d} \|g_{1:T,i}\|_2$$

$$\le \frac{1}{2(1-\beta_1)} \sum_{i=1}^{d} \frac{(\hat{v}_{1,i}^{-1/2}(x_{1,i} - x^*)^2}{\alpha_1} + \frac{\hat{v}_{2,i}^{-1/2}(x_{2,i} - x^*)^2)}{\alpha_2} + \frac{1}{2(1-\beta_1)} \sum_{t=3}^{T} \sum_{i=1}^{d} (x_{t,i} - x_i^*)^2 \left[ \frac{\hat{v}_{t,i}^{-1/2}}{\alpha_t} - \frac{\hat{v}_{t-2,i}^{-1/2}}{\alpha_{t-2}} \right]$$

$$+ \frac{1}{2(1-\beta_1)} \sum_{t=1}^{T} \sum_{i=1}^{d} \frac{\beta_{1t}(x_{t,i} - x^*)^2 \hat{v}_{t,i}^{1/2}}{\alpha_t} + \frac{\alpha\sqrt{1+log(T)}}{(1-\beta_1)^2(1-\gamma)\sqrt{(1-\beta_2)}} \sum_{i=1}^{d} \|g_{1:T,i}\|_2$$

$$\le \frac{D_\infty^2}{2(1-\beta_1)} \sum_{i=1}^{d} \frac{\hat{v}_{1,i}^{-1/2}}{\alpha_1} + \frac{\hat{v}_{2,i}^{-1/2}}{\alpha_2} + \frac{D_\infty^2}{2(1-\beta_1)} \sum_{t=3}^{T} \sum_{i=1}^{d} \left[ \frac{\hat{v}_{t,i}^{-1/2}}{\alpha_t} - \frac{\hat{v}_{t-2,i}^{-1/2}}{\alpha_{t-2}} \right]$$

$$+ \frac{D_\infty^2}{2(1-\beta_1)} \sum_{t=1}^{T} \sum_{i=1}^{d} \frac{\beta_{1t}\hat{v}_{t,i}^{1/2}}{\alpha_t} + \frac{\alpha\sqrt{1+log(T)}}{(1-\beta_1)^2(1-\gamma)\sqrt{(1-\beta_2)}} \sum_{i=1}^{d} \|g_{1:T,i}\|_2$$

$$(12)$$

The third inequality is due to Lemma 2 in Sashank J et al. (2018). The fourth inequality uses the fact that $\beta_{1,t} \le \beta_1$. The fifth inequality uses the property : $\frac{\hat{v}_{t-2,i}^{-1/2}}{\alpha_{t-2}} \le \frac{\hat{v}_{t,i}^{-1/2}}{\alpha_t}$ and the last inequality above uses the following definitions that simplify the notation : $F$ has bounded diameter $D_\infty$ and $g_t = \nabla f_t(\theta_t)$, $\|g_t\|_2 \le G$, $\|g_t\|_\infty \le \|G\|_\infty$ for all $t \in [T]$ and $x \in F$.

Using the telescopic sum, we have the following regret bound :

$$R_{T(2)} \leq \frac{D_\infty^2}{2(1-\beta_1)} \sum_{i=1}^d (\frac{\hat{v}_{T,i}^{-1/2}}{\alpha_T} + \frac{\hat{v}_{2,i}^{-1/2}}{\alpha_2}) + \frac{D_\infty^2}{2(1-\beta_1)} \sum_{t=1}^T \sum_{i=1}^d \frac{\beta_{1t} \hat{v}_{t,i}^{1/2}}{\alpha_t}$$
$$+ \frac{\alpha\sqrt{1+log(T)}}{(1-\beta_1)^2(1-\gamma)\sqrt{(1-\beta_2)}} \sum_{i=1}^d \|g_{1:T,i}\|_2 \tag{13}$$

We can easily see that $R_{T(1)} \leq R_{T(2)}$, because of the term $\frac{\hat{v}_{2,i}^{-1/2}}{\alpha_2} \geq 0$.

Thus, the regret bound of AAMSGrad is :

$$R_T \leq max(R_{T(1)}, R_{T(2)}) = R_{T(2)} \tag{14}$$

Which complete the proof.

