# OpenReview forum: "Accelerating first order optimization algorithms"
_ICLR.cc/2019/Conference_

### Official Review · AnonReviewer2 · 2018-10-30
**Theoretical contribution is limited.**

**Rating:** 4
**Confidence:** 5

**Review:**

Prons:
This paper provides a simple and economic technique to accelerate adaptive stochastic algorithms. The idea is novel and preliminary experiments are encouraging.

Cons:
1.	The theoretical analysis for AAMSGrad is standard and inherits from AMSGrad directly. Meanwhile, the convergence rate of AAMSGrad merely holds for strongly convex online optimization, which does not match the presented experiments. Hence, the theoretical contribution is limited.
2.	The current experiments are too weak to validate the efficacy of the proposed accelerated technique. We recommend the authors to conduct more experiments on various deep neural networks.

---

### Official Review · AnonReviewer1 · 2018-11-02
**Paper is confusing**

**Rating:** 4
**Confidence:** 3

**Review:**

The paper proposes an acceleration method that slightly changes the AMSGrad algorithm when successive stochastic gradients point in different directions.  I found the paper confusing to read because the critical points of Algorithm 1 are very unclear. For instance the \phi function defined by Reddi et al. takes as argument all the past gradients g1...gt (see paper at the bottom of page 3) but is used inside Algorithm 1 with only the current gradient --\phi_t(g_t)-- or an enigmatic "max" of two vectors --\phi_t(max(g_t,pg_t))--  I have no idea what the actual calculation is supposed to be. The proof of the theorem (equation 6 in the appendix) suggests that this is a componentwise maximum and that the other gradients are still in.  But a componentwise maximum is a surprisingly assymetric construction. What if we reparametrize by changing the sign of one particular weight?  We get a different maximum?

I finally looked into the empirical evaluation. I am not sure that the purported effect cannot be ascribed to other factors such as the choice of stepsize --they do not seem to have been looking for the best stepsize for each algorithm. The MNIST experiments are performed with a bizarre variant of CNN that seems to perform substantially worse than comparable system. They show the test loss but not the test accuracy though.

In conclusion I remain confused and unconvinced.

---

### Official Review · AnonReviewer3 · 2018-11-06
**Cannot understand the paper**

**Rating:** 3
**Confidence:** 3

**Review:**

The paper considers a simplistic extension of first order methods typically used for neural network training. Apart from the basic idea the paper's actual algorithm is hard to read because it is full of lacking definitions. I have tried to piece together whatever I could by reading the proof. The algorithm box is very unclear. For instance the * operator is undefined.

To the best of my understanding which the paper changes the update by first checking whether the gradient has the same direction as the previous gradient if yes it uses the component wise maximum of the new gradient and the previous gradient in the update and otherwise it uses the new gradient. Now whether this if condition is checked component wise or an angle between the two vectors is completely unclear.

I will really suggest the authors to at least write their algorithm with clarity. Further while stating the theorem there are undefined parameter and even the objective Regret has not been defined anywhere. Further the theorem which I could not verify due to similar unclarity shows I believe the same convergence result as AMSGrad and hence there is no theoretical advantage for the proposed algorithm. In terms of practice further I do not see a significant advantage and it could result be a step size issue . The authors do not say that they do a search over the hyper parameters.

On a philosophical level it is unclear what the motivation behind this particular change to any algorithm is. It would be good to discuss what additional advantage is added on top of acceleration. Note that the method feels very much like acceleration.

---

### Meta-Review · Area_Chair1 · 2018-12-13
**Issues with the presentation**

**Confidence:** 5
**Recommendation:** Reject

**Metareview:**

Dear authors,

All reviewers commented that the paper had issues with the presentations and the results, making it unsuitable for publication to ICLR. Please address these comments should you decide to resubmit this work.